# 'Off pump' self-expanding injectable tissue valves (IPVR) versus 'on pump' conventional tissue valves (PVR) for replacement of the pulmonary valve: trial protocol for a randomised controlled trial (InVITe trial)

Rachael Heys,[1] Gianni Angelini,[2] Massimo Caputo,[3] Lucy Culliford,[1] Maria Pufulete,[1] Barnaby C Reeves,[1] Chris A Rogers,[1] Serban Stoica,[3] Andrew Parry[2]

[1]Clinical Trials and Evaluation Unit, Bristol Trials Centre, Bristol Medical School, University of Bristol, Bristol, UK
[2]Bristol Heart Institute, University of Bristol, Bristol, UK
[3]Bristol Royal Hospital for Children, University Hospitals Bristol NHS Foundation Trust, Bristol, UK

**Correspondence to**
Andrew Parry;
andrew.parry@uhbristol.nhs.uk

## ABSTRACT

**Introduction** Patients with congenital heart disease often need repeated operations throughout life to replace the pulmonary valve. Valve replacement with 'injectable' self-expanding valves (which is performed 'off pump' without the use of cardiopulmonary bypass, CPB) may result in quicker recovery and lower risk of major complications than valve replacement with conventional valves (which is performed 'on pump' with the use of CPB).

**Methods and analysis** We are conducting a multicentre, single-blind randomised controlled trial in patients with congenital heart disease and aged between 12 and 80 years. We will randomise participants in a 1:1 ratio to receive either 'off pump' injectable pulmonary valve replacement or 'on pump' conventional pulmonary valve replacement. The primary outcome will be the difference between the groups with respect to post-surgery blood loss (as measured by chest drain volume) in the first 24 hours. Secondary outcomes will include in-hospital outcomes (intensive care unit stay, inotropic/vasodilator support, chest drain volume in the first 12 hours post-surgery, time of readiness for extubation, blood products used in the first 24 hours post-surgery, time of fitness for discharge, valve and heart function 6 months post-surgery (assessed using cardiovascular magnetic resonance and ECHOCARDIOGRAPHY) and health-related quality of life 6 weeks and 6 months post-surgery.

**Ethics and dissemination** This trial has been approved by the South West Exeter Research Ethics Committee. Findings will be shared with participating hospitals and disseminated to the academic community through peer reviewed publications and presentation at national and international meetings. Patients will be informed of the results through patient organisations and newsletters to participants.

**Trial registration number** ISRCTN23538073

## Strengths and limitations of this study

► This trial is the first randomised controlled trial (RCT) to assess the effectiveness of injectable tissue valves versus conventional tissue valves for pulmonary valve replacement (PVR).

► This is a multicentre RCT, with computer-generated concealed randomisation and participant blinding to minimise bias.

► We are including both children and adults (age range 12–80 years) to ensure the findings will be generalisable to a wide group of patients who need PVR.

► Although clinicians cannot be blinded to the patient's treatment allocation, we will minimise bias by pre-defining all procedures for data collection and applying these to all participants in the same way.

► The primary and many of the secondary endpoints are clinical outcomes, based on objective criteria, which are important to patients, clinical teams and the NHS.

## INTRODUCTION
### Background and objectives

Most children born with complex congenital heart disease now reach adulthood. Post-surgery mortality following surgical correction, of tetralogy of Fallot, for example, has fallen from 5.5% in 1980[1] to 0% in 2016.[2] However, as long-term survival has improved late sequelae have become evident. Many of the patients operated on for tetralogy of Fallot and other forms of pulmonary valve disease require multiple operations throughout life to repair or replace poorly functioning pulmonary valves.

The standard operation for pulmonary valve replacement (PVR) involves opening

the chest, exposing the heart and using cardiopulmonary bypass (CPB), ('on pump'). CPB is a technique that temporarily takes over the function of the heart and lungs (pumping blood and oxygen through the body) and occasionally involves stopping the heart during surgery. CPB is associated with complications such as excessive bleeding and reperfusion injury in cases where the heart has to be stopped (damage to heart tissue occurs when the blood supply returns to the heart).

Injectable tissue valves (BioPulmonic, Biointegral Surgical) have been developed to replace conventional tissue valves for PVR. Injectable tissue valves can be implanted without the use of CPB (ie, 'off pump') and with less dissection of the heart. Case reports and small case series (<12 patients)[3–10] have shown that injectable tissue valves are easy to implant and have satisfactory function. Pilot data have shown shorter operating time (2 hours less) and less blood loss (over 400 mL less) and blood product requirement (3 units less) in 6 'off-pump' injectable pulmonary valve replacement (IPVR) patients compared with 7 'on pump' PVR patients. No patient had a paravalvular leak (leak around the replaced valve) or more than mild pulmonary regurgitation at early follow-up, suggesting that IPVR is a safe procedure.[11] There are no randomised controlled trials (RCTs) that have evaluated IPVR versus PVR.

We are conducting an RCT to determine whether IPVR leads to less blood loss, faster recovery, lower risk of post-surgery complications and good valve function in patients who need an operation to replace their pulmonary valve. We could not consider percutaneous transcatheter pulmonary valve replacement (TPVR), which is performed via cardiac catheterisation and therefore does not require open-heart surgery, because current TPVR valve sizes are only suitable for patients with a right ventricular outflow tract (RVOT) diameter of ≤27 mm which would exclude >80% of our eligible population.[12 13] Further, all eligible patients will have native outflow tracts that are too distensile for robust fixation of a percutaneous valve.

## AIMS AND OBJECTIVES
The main aim of the InVITe trial is to compare IPVR versus PVR in patients who need PVR. The objective is to estimate the difference between the groups with respect to:
1. Post-surgery blood loss (as measured by chest drain volume) in the first 24 hours (primary outcome);
2. A range of clinical post-surgery outcomes (intensive care unit (ICU) stay, inotropic/vasodilator support, chest drain volume in the first 12 hours, time until ready for extubation, blood products used in the first 24 hours, fitness for discharge (FFD) and valve-related complications during follow-up);
3. Heart and valve function assessed using cardiovascular magnetic resonance (CMR) and Echocardiography;

4. Health related quality of life (HRQoL).

## METHODS
### Trial design and population
The InVITe trial is an early phase, multicentre, RCT comparing IPVR with PVR. We will recruit patients being referred for PVR at University Hospitals Bristol NHS Foundation Trust and University Hospitals Southampton NHS Foundation Trust.

### Eligibility criteria
Potential participants will be eligible if they are:
1. Aged between 12 and 80 years.
2. Have a valve size between 25–31 mm.
3. Undergoing any of the following as either a first or a redo procedure:
   a. Replacement of the pulmonary valve;
   b. Replacement of the pulmonary valve with atrial septal defect which is amenable to closure via cardiac catheter or;
   c. Replacement of the pulmonary valve with right ventricular outflow tract (RVOT) reconstruction that does not require CPB if using an injectable valve.

Potential patients will be ineligible if they are:
1. Prisoners or adults lacking capacity to consent.
2. Aged between 12 and 15 years and under the care of social services.
3. Undergoing an additional procedure that requires the use of CPB (eg, pulmonary valve or artery stenosis requiring patch reconstruction, intra-cardiac shunt, RVOT reconstruction requiring CPB, other anatomical heart corrections).
4. Diagnosed with active endocarditis.
5. Unwilling to undergo surgery involving a porcine product.
6. Unable to assent/consent.

### Patient approach and consent
We will identify potential participants from clinic lists (elective patients) and theatre schedules (urgent patients) and they will be given an invitation letter and patient information leaflet (PIL) suitable to their age group. Parent(s)/guardian(s) of potential participants 12–15 years old will be given an invitation letter and parent/guardian information leaflet (P/GIL). Most patients and parent(s)/guardian(s) will have at least 24 hours to consider whether they would like to participate, although in some cases this time interval may be shorter; for example, patients admitted from other hospitals without prior notification to the research team. Written informed consent/assent will be obtained from the patients and/or their parent(s)/guardian(s) (where appropriate) prior to inclusion in the trial. Figure 1 shows the patient pathway.

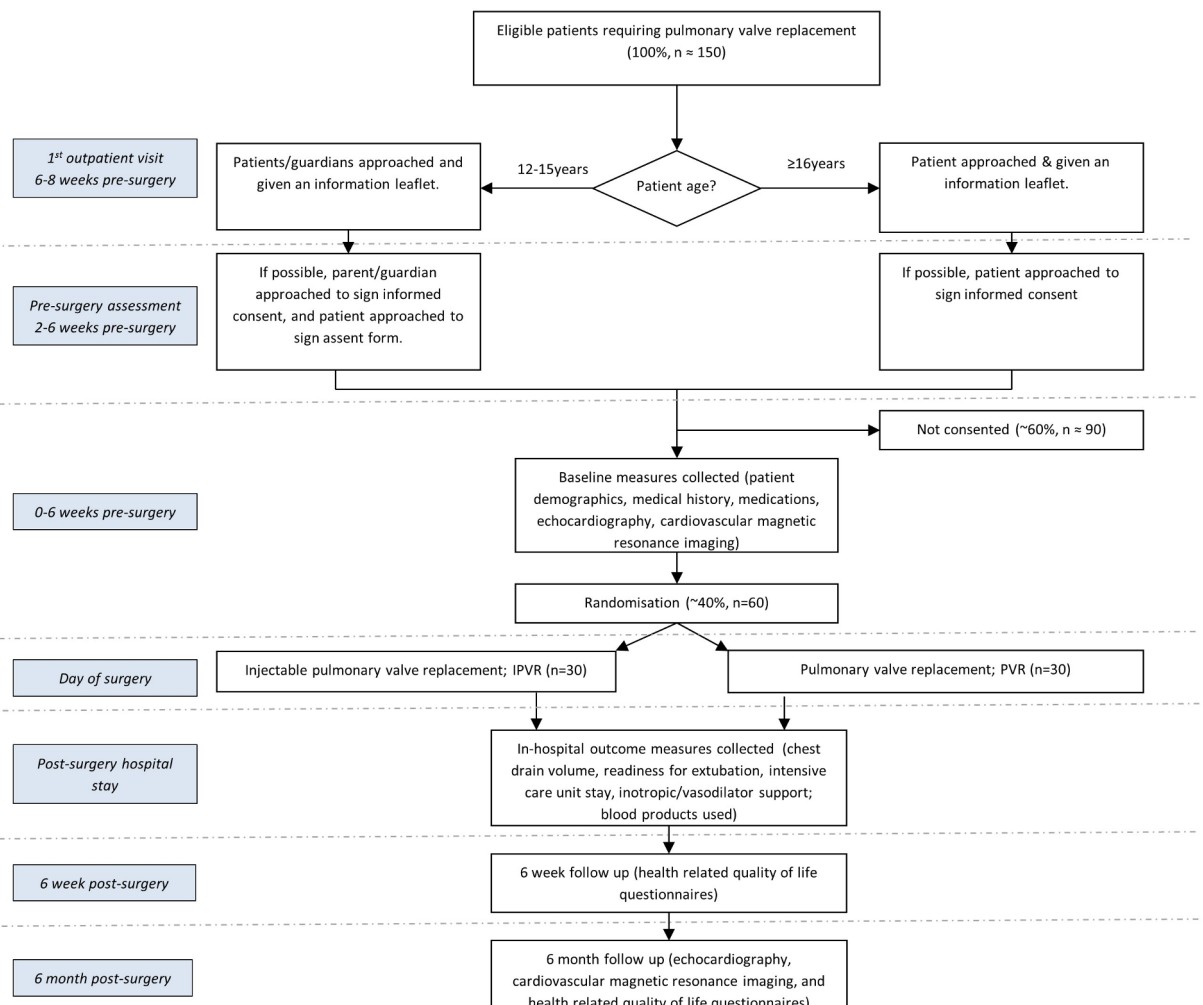

**Figure 1** Trial schema showing the recruitment pathway with the number of patients to be recruited, anticipated eligibility and recruitment rates.

## Trial interventions

For both operations the heart will be exposed through a standard incision for a heart operation (median sternotomy).

For IPVR operations a self-expanding no-react injectable BioPulmonic valve (Biointegral Surgical) will be used. The main pulmonary artery and the front of the heart will be dissected out. A purse-string stitch will be placed in the front of the heart and the size of valve required will be confirmed using an echocardiogram. If the main pulmonary artery is too big, it will be made smaller using a stitch to narrow it down. The valve injector will be inserted into the heart through the purse string and the valve will be deployed and sewn into position using 3 fixation stitches. If the patient becomes unstable during the surgery, CPB will be instituted. The position and function of the valve will be confirmed using transoesophageal echocardiography. The operation will then be completed in the standard manner.

For PVR operations, the type of tissue valve used will be according to the usual practice at the participating site and may include stented tissue valves (in adults particularly) or homografts (in children particularly). The heart will be freed from the scar tissue and the patient put on CPB. The pulmonary artery will be opened and the old valve removed. A new valve of the correct size will be sewn into place. In some cases the heart may need to be stopped for a short period of time. A patch may also be necessary to close the heart over the new valve. Once this has been completed and the heart has recovered from the effects of cardioplegic arrest, CPB will be discontinued and the operation be completed in the standard manner.

All other aspects of participant's care will be performed according to standard practice.

Participants will be followed up at 6 weeks (questionnaire completion) and 6 months (questionnaire completion, Echocardiography and CMR) post-surgery. The 6-month Echocardiography and CMR will coincide with routine hospital follow-up; so we expect minimal loss to follow-up. A 6-month follow-up period will provide adequate data to inform early recovery.

## Randomisation

Trial allocations will be generated by a computer using block randomisation with varying block sizes. Participants

will be randomly assigned in a 1:1 ratio and randomisation will be stratified by centre. If a participant's operation is unexpectedly rescheduled, the trial number and randomised allocation will be retained.

### Trial outcomes

The primary outcome will be post-surgery blood loss as measured by the chest drainage volume in the first 24 post-surgical hours. The secondary outcomes include:

1. Post-surgery time to 'readiness for extubation'. The following criteria will be documented at the time of extubation (when all criteria should be met): normal temperature, cardiovascular system stable, no metabolic imbalance, blood loss decreasing and below a defined rate, able to clear respiratory secretions, patient awake, no residual muscle paralysis and adequate analgesia;
2. Length of ICU stay;
3. Inotropic/vasodilator support;
4. Chest drain volume in the first 12 post-surgical hours;
5. Blood products used in the first 24 post-surgical hours (red blood cells, fresh frozen plasma, platelets, human albumin solution, cryoprecipitate) and bleeding events that may or may not lead to re-operation;
6. Post-surgery time to FFD, defined as when all of the following criteria are met: normal temperature, pulse and respiration, normal oxygen saturation on air, normal bowel function, physically mobile ;
7. CMR assessment of valve and heart function at 6 months, defined as presence and degree of pulmonary regurgitation, end-diastolic volume and right ventricular ejection fraction;
8. Echocardiographic assessment of valve function at routine follow-up at 6 months, defined as presence and degree of pulmonary regurgitation and residual valve stenosis, measured as peak velocity through the valve;
9. HRQoL at 6 weeks and 6 months post-surgery (EuroQol EQ-5D in patients ≥18 years and EQ-5D-Y in patients <18 years, SF36 in patients ≥18 years and child health questionnaire in patients <18 years);
10. Valve-related complications up to 6 months post-surgery.

### Data collection

The schedule of data collection is shown in table 1. Data will be collected from medical notes and hospital records, entered onto a bespoke database and stored on a secure server. Access to the database will be limited to authorised personnel. Data will be collected and retained in accordance with the UK Data Protection Act 2018.

### Blinding

We will blind participants to their treatment allocation and assess the efficacy of blinding at 6 weeks and 6 months post-surgery via questionnaires. We will record any instances of unblinding. Staff responsible for the care of the patient will be aware that the patient is blinded and asked not to disclose the patient's treatment allocation. It will not be possible to blind surgeons, staff responsible for the care of the patient or research nurses

**Table 1** Schedule of data collection

| Data item | Pre-surgery | During surgery | Post-surgery (up to and including discharge) | 6 weeks post-surgery | 6 months post-surgery |
|---|---|---|---|---|---|
| Screening and consent data | ✓ | | | | |
| Baseline data (patient demographics, medical history, medications) | ✓ | | | | |
| Cardiovascular magnetic resonance imaging | ✓ | | | | ✓ |
| Echocardiography | ✓ | | | | ✓ |
| Randomisation allocation | ✓ | | | | |
| Intervention (injectable pulmonary valve replacement /PVR) | | ✓ | | | |
| Inotropic/vasodilator support | | | ✓ | | |
| Chest drain volume in the first 12 and 24 hours | | | ✓ | | |
| Blood products used in the first 24 hours | | | ✓ | | |
| Readiness for extubation | | | ✓ | | |
| Intensive care unit stay | | | ✓ | | |
| Fitness for discharge | | | ✓ | | |
| Health related quality of life | | | | ✓ | ✓ |
| Follow-up questionnaires to collect all serious post-surgery complications and safety data | | | | ✓ | ✓ |

to the participant's treatment allocation. Details of the procedure are needed to order an injectable valve of the required size and also need to be recorded in the medical notes.

## Sample size calculation

A sample size of 60 (30 per group) will have 80% power to detect an approximate 50% reduction in chest drainage volume (effect size on logarithmic scale of −0.73) over the first 24 hours at a 5% significance level (two-tailed). This target difference is large but is entirely consistent with the >70% reduction observed in a small non-randomised comparison of patients who had IPVR and PVR.[11]

## Statistical analysis

The trial will be analysed on an intention-to-treat basis, that is, outcomes will be analysed according to the treatment allocation. Any non-adherence to the allocated group will be documented. The primary outcome (chest drainage volume in the first 24 post-surgical hours) will be analysed using linear regression (on the logarithmically transformed values as the distribution is expected to be skewed). All secondary outcomes will yield either binary, quantitative, time-to-event or longitudinal data, and will be analysed using logistic regression (binary outcomes), linear regression (quantitative outcomes), survival methods (time-to-event outcomes). Outcomes with repeated measures (longitudinal data) will be analysed using mixed models which allow for unbalanced data. Alternative correlation structures will be considered and the sensitivity of the results to the choice of the structure examined. Analyses will be adjusted for the centre (stratification factor). Outcomes will be reported as effect sizes with 95% CIs.

## Patient and public involvement

Patients were not actively involved in the design of the study. However, the PIL for patients aged 12–15 years was reviewed by three children and their families (the children were members of the public who did not have congenital heart disease) to confirm the acceptability of the study and the content and format of the PIL as being appropriate for this age group. Changes to the PIL were made based on their feedback to ensure the PIL was suitable for this age group.

Patients will be informed of the results through patient organisations and through a summary report for participants. The results summary for participants will be designed with input from the Cardiovascular Biomedical Research Centre patient public involvement advisory group.

## Risk of bias

We have designed the trial to minimise the risk of bias. Participants will be randomised after eligibility is confirmed. Participants will be blinded to treatment allocation to ensure that patient-reported outcomes (HRQoL) are not influenced by any perceptions about which valve is better.

Although surgeons, staff responsible for the care of the patient and research nurses will not be blinded to the treatment allocation, all surgical procedures will follow standard protocols. We will pre-define all procedures and data collection required for participant follow-up and apply the procedures to all participants in the same way.

We expect minimal missing outcome data as most outcome data will be collected during the participants' hospital stay. We will maintain contact with participants during follow-up (eg, contacting patients with overdue questionnaires) to maximise questionnaire completion rates and attendance at the 6-month follow-up appointment.

We have pre-specified all outcomes and will write a detailed analysis plan before the database is closed and comparisons between groups are investigated to prevent selective reporting of results.

## ETHICS

The trial is managed by the Clinical Trials and Evaluation Unit Bristol (CTEU Bristol) and sponsored by University Hospitals Bristol NHS Foundation Trust (www.uhbristol.nhs.uk/research-innovation/). Participants and their parents/guardians (where applicable) have the right to withdraw at any time and if they do withdraw, will be treated according to their hospitals' standard procedures. Participants who choose to withdraw from the study will be asked if we can retain and continue to use any data already collected and whether they are willing to participate in the trial follow-up.

### Changes to the protocol since REC approval

We made three major changes to the protocol before recruitment began. First, we decided to blind participants to ensure validity of the patient-reported outcome data. Second, we decided that it was not feasible to blind the research nurses to the treatment allocation as they would be required to order the valves. Third, initially we intended to establish a Data Monitoring and Safety Committee to oversee the trial. Given that the trial has a relatively low target sample size, a short follow-up period and we expect very few serious adverse events, it is hard to conceive of an interim analysis (eg, after 50% of the participants have been recruited) with sufficient power to identify a safety issue that would require action to be taken. We concluded, therefore, that it was sufficient for the safety data to be reviewed by the Trial Management Group and Trial Steering Committee for Cardiovascular Studies, a sub-group of the University Hospitals Bristol NHS Foundation Trust Cardiovascular Research Board. Shortly after opening to recruitment, we made one major change to the protocol, to extend the baseline data collection to include the pre-surgery echocardiographic and CMR results. Finally, protocol and study documents have been amended to allow the central analysis of CMR data. Version 6.0 (dated 02/07/2018) of the protocol is currently in use. The relevant regulatory approvals will

be obtained for amendments to the protocol and study information leaflets. Relevant parties (eg, investigators at participating sites) will be informed via email. If changes are made to the study processes, participants consent will be obtained again, if necessary.

### Recruitment and issues experienced

The InVITe trial opened to recruitment in April 2016 in Bristol and the first patient was recruited in July 2016. Recruitment in Southampton opened in January 2017. To date, we have identified 38 eligible patients and 17 patients have consented, 13 of whom have had surgery (8 of these have completed follow-up).

Recruitment has been slower than anticipated for a number of reasons. Several sites were approached to participate in the study but declined. Reasons given included a reluctance to depart from the traditional care pathway for high-risk patients, excessive treatment costs involved and the motivation of surgeons to undertake the study. In the recruiting sites, the number of patients being referred for a PVR is lower than expected and the number of patients who do not meet the eligibility criteria is higher than originally anticipated.

Additionally, organising training in IPVR surgery has proved logistically challenging. As part of the trial, surgeons were required to complete two training cases which were supervised by a leading expert in IPVR surgery. It took longer than expected for all participating surgeons to complete the required training. During this time, patients whose operations were scheduled at short notice could not be approached. In Bristol, there were initially two participating surgeons. To help improve recruitment rates, we invited an additional surgeon to participate in the study.

## DISSEMINATION OF FINDINGS

We will present our findings at international meetings and in peer-reviewed publications. We will inform the public through patient organisations and through newsletters to participants.

## DISCUSSION

The InVITe trial will be the first RCT to compare IPVR with PVR in patients requiring replacement of the pulmonary valve. To our knowledge, the trial remains the only RCT (published or registered) comparing IPVR with conventional PVR. The proposed trial should, therefore, contribute significantly to the understanding of IPVR and inform larger RCTs in the future.

Most individuals with congenital heart disease need repeated operations to replace the pulmonary valve. For example, if a child needs a pulmonary valve replaced, it is more likely that the prosthetic valve will need to be replaced more than once as the tissue valve will wear out. It is therefore essential to look at ways to minimise harm and improve the quality of life after surgery. The main benefits for patients due to IPVR are believed to be a quicker recovery as well as a lower risk of post-surgery morbidy, including serious but rare (<5%) complications. These benefits to patients are expected to translate into benefits for the NHS with fewer resources needed during and after the operation (shorter surgery time and no CPB costs, shorter duration of ventilation, intensive care and overall hospital stay, fewer blood products and fewer resources required to treat comorbidity).

The main strengths of our trial are the inclusion of both children and adults which will ensure that our findings are generalizable to a wide range of patients, the collection of data on a wide range of primary and secondary outcomes which could inform the selection of outcomes in a future larger RCT and the inclusion of a patient-reported outcome (HRQoL) which is important in a population that needs repeated operations.

**Acknowledgements** The InVITe trial is sponsored by University Hospitals Bristol NHS Foundation Trust. The sponsor is responsible for the oversight of the InVITe study and will ensure the trial is managed appropriately. This trial was designed and delivered in collaboration with the Clinical Trials and Evaluation Unit (CTEU), a UKCRC registered clinical trials unit which, as part of the Bristol Trials Centre is in receipt of National Institute for Health Research CTU support funding. The research team acknowledges the support of the National Institute for Health Research Clinical Research Network (NIHR CRN). The authors would like to thank all trial team members involved in the recruitment, coordination and data entry for this trial and members of the CV-BRC PPI group and public who assisted with the review of study documents. Special thanks are also given to perfusion, anaesthetics and nursing teams for their support in the conduct of the trial and to Pierson Ltd and Stefano Marianeschi for their help in providing IPVR training.

**Contributors** RH: Preparation and drafting of trial protocol, writing manuscript. GA: Trial design, review of trial protocol, review of manuscript. MC: Participating surgeon in the trial, review of manuscript. LC: Trial design, preparation and drafting of trial protocol, review of manuscript. MP: Preparation and drafting of trial protocol, review of manuscript. BCR: Trial design, preparation and drafting of trial protocol, review of manuscript. CAR: Trial design, preparation and drafting of trial protocol, sample size and statistical analysis plan, review of manuscript. SS: Participating surgeon in the trial, review of manuscript. AP: Trial concept, study design, preparation of trial protocol, definition of trial interventions, definition of outcomes, writing manuscript. Chief Investigator. All authors read and approved the final manuscript.

**Funding** The trial is funded by the National Institute for Health Research (NIHR) Bristol Cardiovascular Biomedical Research Centre (Cardiovascular theme) and the British Heart Foundation.

**Competing interests** Pierson Ltd have agreed to provide the injectable tissue valves (BioPulmonic™, Biointegral Surgical Inc.) used in the trial at a discounted rate. Pierson Ltd paid for the expenses associated with Stefano Marianeschi to visit the trial centres and provide IPVR training.

**Patient consent for publication** Not required.

**Ethics approval** The trial received research ethics approval from the South West Exeter Research Ethics Committee (REC) in November 2015.

**Provenance and peer review** Not commissioned; externally peer reviewed.

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
