## [Reviewer comments · BMJ Open]

This paper was submitted to a another journal from BMJ but declined for publication following peer review. The authors addressed the reviewers' comments and submitted the revised paper to BMJ Open. The paper was subsequently accepted for publication at BMJ Open.

(This paper received three reviews from its previous journal but only two reviewers agreed to published their review.)

ARTICLE DETAILS

TITLE (PROVISIONAL)	'Off pump' self-expanding injectable tissue valves (IPVR) versus 'on pump' conventional tissue valves (PVR) for replacement of the pulmonary valve: trial protocol for a randomised controlled trial (InVITe trial)
AUTHORS	Heys, Rachael; Angelini, Gianni; Caputo, Massimo; Culliford, Lucy; Pufulete, Maria; Reeves, Barnaby; Rogers, Chris; Stoica, Serban; Parry, Andrew

VERSION 1 – REVIEW

REVIEWER	Chan, Kwan leung University of Ottawa Heart Institute, Canada
REVIEW RETURNED	25-Oct-2018

GENERAL COMMENTS	This is a study to compare an injectable valve implanted off pump with a porcine valve implanted on pump in pulmonary replacement in patients with congenital heart disease. The primary objective is blood loss from chest drain in the first 24 hours. There needs to be a discussion of other options particularly in relation to percutaneous implantable valves which are gaining in popularity and may obviate the demand for the injectable valve that is being studied. It may be more pertinent to have 3 arms in the study to include a group with the percutaneous option. In the current form, the impact of the study may be limited.
--

REVIEWER	Ioanna Kosmidou Columbia University
REVIEW RETURNED	31-Oct-2018

GENERAL COMMENTS	The protocol presented by Heys and colleagues describes an early phase randomized comparison of injectable PV off CPB vs conventional PVR on CPB. The topic is of interest and addresses an issue of increasing frequency among the CHD patient population, given improving longevity. Nevertheless, there are several major concerns to be addressed: 1. The authors do not clearly state anywhere that they are using a specific type of IPV; this should be clarified in the text. Further, there is no mention of the type(s) of PV used in the conventional PVR arm. 2. The choice of primary outcome is interesting; the authors present the study as an efficacy study, yet the primary endpoint of
--

	post surgical blood loss is certainly not an efficacy endpoint. Further, why is post surgical blood loss (than surgical and post surgical) used and how are other bleeding events going to be interpreted (for example, catheter related bleeds or spontaneous bleeds)? Which blood products will be included in analysis of the secondary endpoint of blood product use (PRBC or any blood products)? Similarly, it is unclear which post surgical complications and outcomes are to be collected and at what time intervals; this is especially important if the authors wish to present this as an efficacy study. 3. What is the rationale for discontinuing follow up at 6 months? Is there a plan for event adjudication and if not, are clinical outcomes merely reported by site investigators? Please add. 4. The patient population, including inclusion and exclusion criteria, is poorly defined. For example, are redo procedures allowed? Are there any limitations as to RV or LV function for inclusion? How are patients with a history of bleeding diathesis or immunosuppression handled? 5. The authors suggest cost-related limitations partially explaining slow recruitment; do they plan on performing a cost effectiveness analysis? It appears implausible that an injectable valve - provided at a lower cost - which allows for avoidance of CPB is associated with a higher cost. Please address. 6. Please discuss the role of the scientific steering committee in reviewing patient eligibility
--	--

VERSION 1 – AUTHOR RESPONSE

Reviewer: 1

1. This is a study to compare an injectable valve implanted off pump with a porcine valve implanted on pump in pulmonary replacement in patients with congenital heart disease. The primary objective is blood loss from chest drain in the first 24 hours. There needs to be a discussion of other options particularly in relation to percutaneous implantable valves which are gaining in popularity and may obviate the demand for the injectable valve that is being studied. It may be more pertinent to have 3 arms in the study to include a group with the percutaneous option. In the current form, the impact of the study may be limited.

We agree with the reviewer that an RCT including transcatheter pulmonary valve replacement (TVPR) is warranted; however, TVPR would not be a suitable third group in our study for a variety of reasons. First, because of the size limit on TVPR valves, >80% of our PVR population would be excluded which would make the trial infeasible; we have added a statement to this effect at the end of the Introduction (page 5). Second, the patients selected all have native outflow tract tissue, not conduits. The native outflow tracts are highly distensible and not suitable for percutaneous valve implantation as the valve cannot be fixed in place; this is the great advantage of the IPPV. We are aware that pretesting is becoming an option, but this is not currently strongly supported by interventional clinicians in our institution.

Reviewer: 2

1. The authors do not clearly state anywhere that they are using a specific type of IPV; this should be clarified in the text. Further, there is no mention of the type(s) of PV used in the conventional PVR arm

The type of injectable pulmonary valve that will be used in the study has been added to the 'Trial Interventions' section on page 7 of the manuscript. We have not specified the type of tissue valve to be used for standard PVR; this will be the type of valve that is used as part of usual care at participating sites. We have added a sentence stating this in the 'Trial Interventions' section on page 7 of the manuscript. We will collect data on the type of pulmonary tissue valve used.

2. The choice of primary outcome is interesting; the authors present the study as an efficacy study, yet the primary endpoint of post-surgical blood loss is certainly not an efficacy endpoint.

Post-surgical blood loss is a relevant outcome to the study because IPVR avoids the use of cardiopulmonary bypass (CPB) and CPB is believed to increase post-operative bleeding. Previous experience of a small case series comparing IPVR (6 patients) with PVR (7 patients) showed that patients with IPVR had less post-operative blood loss.(1) Furthermore, this outcome is objective and allows us to detect a clinically significant difference with a relatively small sample size, which is important given the relatively small number of eligible participants available for recruitment. Other measures of efficacy, such as time to extubation or ICU stay, are very subjective and influenced by many other factors other than clinical condition.

3. Further, why is post-surgical blood loss (than surgical and post-surgical) used and how are other bleeding events going to be interpreted (for example, catheter related bleeds or spontaneous bleeds)? Which blood products will be included in analysis of the secondary endpoint of blood product use (PRBC or any blood products)?

We will collect data on all bleeding events from the end of surgery to 24 hrs post-operatively. Intraoperative blood loss is not related to the valve type used. Furthermore, it is difficult to accurately measure. Therefore, we do not think it is relevant in demonstrating a difference between the two groups.

Secondary endpoints will include the following; transfusion of red blood cells; fresh frozen plasma, FFP; platelets; cryoprecipitate; human albumin solution. The section 'Trial outcomes' on page 9 of the manuscript has been updated to include the blood product data that will be collected.

4. Similarly, it is unclear which post-surgical complications and outcomes are to be collected and at what time intervals; this is especially important if the authors wish to present this as an efficacy study.

We will collect all serious post-surgical complications that occur during the period from surgery to 6 months post-surgery (e.g. cardiac arrest, pericardial effusion and re-operation etc). 'Table 1 Schedule of data collection' on page 10 of the manuscript has been updated with this information.

5. What is the rationale for discontinuing follow up at 6 months? Is there a plan for event adjudication and if not, are clinical outcomes merely reported by site investigators? Please add.

There is no plan for event adjudication. The clinical outcomes will be retrieved from patient's clinical notes by the research nurse teams at participating sites. The primary outcome is objective (chest drain volume), as are most of the secondary outcomes.

Since the main aim of the study is to assess recovery after surgery, 6 months is an adequate time frame to inform the study outcomes. The valves (both standard and injectable) have a longevity of several years; although indefinite follow-up is possible using routine data sources, extended followup is not planned in the current protocol. Any early failures of the valve replacement are likely to occur within the study follow up period and will be recorded as a complication. This time point also coincides with when patients normally attend hospital for follow up after a pulmonary valve replacement and are discharged. This will ensure that loss to follow up is minimal.

6. The patient population, including inclusion and exclusion criteria, is poorly defined. For example, are redo procedures allowed? Are there any limitations as to RV or LV function for inclusion? How are patients with a history of bleeding diathesis or immunosuppression handled?

All patients undergoing either a first time or re-do pulmonary valve replacement will be eligible to participate. There are no additional exclusion criteria beyond the criteria already listed in the

manuscript. However, we will be collecting information at baseline on all the variables highlighted by the reviewer. In all other respects, patients are managed in an entirely standard manner.

7. The authors suggest cost-related limitations partially explaining slow recruitment; do they plan on performing a cost effectiveness analysis? It appears implausible that an injectable valve - provided at a lower cost - which allows for avoidance of CPB is associated with a higher cost. Please address.

As this is a relatively small study, we do not plan to conduct a cost-effectiveness analysis. The discounted cost applies only to the study, i.e. to make the study feasible for participating hospitals when the injectable valve is not part of current routine care. Data collection in the study will capture the most expensive resources used and would, in principle, allow a simple cost-minimisation analysis to be carried out.

8. Please discuss the role of the scientific steering committee in reviewing patient eligibility

The Trial Steering Committee will review the number of patients screened, eligibility rates and recruitment rates. The committee will review the reasons for ineligibility. They can suggest alterations to the eligibility criteria if they think this would be beneficial for the study.

References

1. Chen Q, Turner M, Caputo M, Stoica S, Marianeschi S, Parry A. Pulmonary valve implantation using self-expanding tissue valve without cardiopulmonary bypass reduces operation time and blood product use. *J Thorac Cardiovasc Surg.* 2013;145(4):1040-5.

VERSION 2 – REVIEW

REVIEWER	Ioanna Kosmidou Columbia University USA
REVIEW RETURNED	28-Nov-2018
GENERAL COMMENTS	The authors addressed comments from R1 adequately.